# IGF-1 Stimulates Glycolytic ATP Production in MCF-7L Cells

**DOI:** 10.3390/ijms241210209

**Published:** 2023-06-16

**Authors:** Bhumika Rajoria, Xihong Zhang, Douglas Yee

**Affiliations:** 1Department of Pharmacology, University of Minnesota, Minneapolis, MN 55455, USA; bhumi003@umn.edu; 2Masonic Cancer Center, University of Minnesota, Minneapolis, MN 55455, USA; zhang004@umn.edu; 3Department of Medicine, University of Minnesota, Minneapolis, MN 55455, USA

**Keywords:** type 1 insulin-like growth factor receptor, insulin receptor, type 1 insulin-like growth factor, insulin, metabolic dysregulation, breast cancer

## Abstract

The Insulin-like Growth Factor (IGF) system in breast cancer progression has been a matter of interest for decades, but targeting this system did not result in a successful clinical strategy. The system’s complexity and homology of its two receptors—insulin receptor (IR) and type 1 insulin-like growth factor receptor (IGF-1R)—are possible causes. The IGF system maintains cell proliferation and also regulates metabolism, making it a pathway to explore. To understand the metabolic phenotype of breast cancer cells, we quantified their real-time ATP production rate upon acute stimulation with ligands—insulin-like growth factor 1 (1GF-1) and insulin. MCF-7L cells express both IGF-1R and IR, while tamoxifen-resistant MCF-7L (MCF-7L TamR) cells have downregulated IGF-1R with unchanged IR levels. Treating MCF-7L cells with 5 nM IGF-1 increased the glycolytic ATP production rate, while 10 nM insulin did not affect metabolism when compared with the control. Neither treatment altered ATP production in MCF-7L TamR cells. This study provides evidence of the relationship between metabolic dysfunction, cancer, and the IGF axis. In these cells, IGF-1R, and not IR, regulates ATP production.

## 1. Introduction

Breast cancer in the United States has the second most common occurrence after skin cancer, with 13% odds of a woman developing it in their lifetime [1]. With a mortality rate of 2.5%, breast cancer is only second to lung cancer as the leading cause of death in women [1]. Projections for 2023 indicate that about 297,790 women will be diagnosed with invasive breast cancer, and 43,700 of them will succumb to the disease [1]. Increased awareness, earlier screening, and better treatments have led to a steady decline in the rate of death since 1989. Moreover, an increase in incidence rates each year makes understanding the underlying systems and their mechanisms to develop additional targeted therapies for cancer ever so important.

One such system—the insulin-like growth factor (IGF) system—is critical for cell survival since it regulates cell growth, proliferation, and metabolism [2,3]. It has also been identified as a prerequisite for various cancers and plays a role in breast cancer development as well [4,5]. Growth factors assist cancer cells in sustaining proliferation by deregulating the signaling capacity [5]. High levels of circulating type 1 IGF (IGF-1) and hyperinsulinemia have been associated with higher breast cancer risk and poor prognosis [6]. IGF studies have indicated oncogene transformation, tumor proliferation, and tumor growth regulation by type 1 IGF receptor (IGF-1R), IGF-1, and insulin [7], identifying IGF signaling a viable therapeutic target.

### 1.1. The IGF System Structure and Signaling

The IGF family comprises several components (Figure 1), including transmembrane tyrosine kinase receptors (RTKs)—IGF-1R, type 2 IGF receptor (IGF-2R), insulin receptor (IR) A and B, and their hybrids; three structurally and functionally homologous ligands—IGF-1, IGF-2, and insulin; and six soluble binding proteins: IGF binding proteins (IGFBPs) 1–6 [8]. IGF-1R and IR receptors are structurally similar tyrosine kinase receptors with more than 50% and 80% homology in sequence and intracellular kinase domains [9]. The homology is attributed to a duplication event on their common ancestral gene. Two α and β subunits are synthesized from a single mRNA precursor, which undergoes glycosylation, proteolytic cleavage, and disulfide bonding to form a functional transmembrane αβ dimer [10]. The dimer—also referred to as a “half receptor”—has an extracellular ligand domain on its α chain and a tyrosine kinase catalytic domain on the intracellular β chain [11]. Although these receptors belong to the RTKs family, they differ from others as covalent dimers, constituting two α and β subunits and requiring ligand binding to dimerize. IR is believed to have a high affinity for insulin and 10-fold and 50–100 fold lower affinities for IGF-1 and IGF-2, respectively [12,13]. However, IR-A, an IR isoform, has been shown to have a higher affinity for insulin as well as IGF-2 as compared to IGF-1 [14,15,16]. On the other hand, IGF-1R has a high affinity for IGF-1 and IGF-2 ligands and a 500–1000 lower affinity for insulin, confirming insulin’s growth factor properties at high concentrations [12].

Ligand binding results in the dimerization of these half receptors to form functional holo-receptors as well as hybrid receptors with IR and IGF-1R halves. Ligand binding also causes conformational changes resulting in the induction of intracellular tyrosine kinase activity and transphosphorylation of the β subunit. Adaptor proteins, including insulin receptor substrates (IRS) (1–6) and Src-homology collagen (SHC), are recruited and phosphorylated by the activated receptor, which stimulates various downstream signaling pathways, including phosphoinositide-3-kinase-protein kinase B/AKT (PI3K-AKT) and RAS-RAF-Mitogen activated protein kinase (MAPK)-ERK. These then regulate proliferation, survival, metabolism, metastasis, apoptosis, and angiogenesis [17,18].

Circulating IGF-1 is mainly produced in the liver along with most extrahepatic tissues, such as the brain, or neoplastic tissue. It is an endocrine, paracrine, and autocrine growth stimulator that maintains physiological processes or tumor growth [18,19]. IGF-1 also has insulin-like metabolic effects, such as elevating glucose uptake and inducing hypoglycemia without lowering fatty acid levels, and interacts with IGF-1R primarily to elicit these responses [8,20,21]. IGF-2 has implications for embryonic growth and is expressed by the liver in adults along with most tissues [22,23]. Glucose, protein, and lipid metabolism is regulated by insulin and IR interaction through PI3K/AKT and RAS/RAF/MAPK/ERK cascades [24]. Metabolic alteration is a hallmark of cancer, and RTK signaling dysregulation may be implicated in metabolic reprogramming occurring in breast cancers [25,26].

### 1.2. IGF Axis in Metabolism and Breast Cancer Risk

Glucose metabolism preferentially occurs via glycolysis rather than oxidative phosphorylation in cancer cells, as described by Warburg [27]. ATP generation via glycolysis is less efficient compared to oxidative phosphorylation and thus necessitates high glucose uptake in cancer cells [28,29]. However, glycolysis is advantageous in promoting cell proliferation by providing substrates for nucleic acid and fatty acid synthesis [26]. Over recent years, identifying the correlation between host metabolism and cancer has gained more importance. Preclinical studies have implicated insulin in tumor promotion by either directly or indirectly affecting epithelial tissues with the help of IGFs [30]. Hyperinsulinemia has strong anabolic effects leading to proliferative tissue abnormalities, DNA synthesis, and cell proliferation, which strongly suggests insulin’s role in cancer progression [31]. A higher concentration of IR in breast cancer (BC) cells vs. normal breast tissue has also been reported [32], leading to the questioning of its role in breast cancer development and investigating its potential as a therapeutic target. Further, epidemiologic studies show evidence of the correlation between high levels of plasmatic IGFs and increased risk of several cancers. It has also been reported that loss of tumor suppressor genes such as BRCA1, p53, and PTEN leads to an increase in IGF-1R expression [33]. In postmenopausal ER+ and PR+ breast cancer patients, obesity has been associated with breast cancer progression [34,35]. Cell proliferation has been linked to metabolic syndrome, characterized by increased BMI and circulating insulin and growth factors, suggesting their mutual involvement in breast cancer progression.

### 1.3. Targeting the IGF System

Strategies to target this system focus on preventing downstream signaling activation by various approaches, including disrupting ligand–receptor interaction, inhibiting tyrosine kinase domain of receptors, and neutralizing ligand activation [18]. Monoclonal antibodies (mAb) against IGF-1R, tyrosine kinase inhibitors (TKIs) against IGF-1R and IR, and small peptide inhibitors against IGF-1R and IR showed success in either in vitro or in vivo or both [36,37,38,39,40]. However, they had no clinical activity in phase 3 trials in numerous cancers and caused the side effects of hyperglycemia [41] along with hyperinsulinemia [42,43] due to IGF-1R targeting. Studies attribute this failure to an increase in IR function after IGF-1R loss in endocrine-resistant BC cells [44,45]. Devising a treatment system targeting both IGF-1R and IR might help avoid the compensatory mechanism. However, IR targeting raises concerns about deregulated glucose metabolism and thus stems the need for studying the overlap in functions of IGF-1R and IR more extensively. Moreover, cancer cells hold the advantageous capacity of metabolic switching, which contributes to therapy resistance and immunity from drug toxicity [46].

Understanding the metabolic phenotype of BC cells can help identify useful trends for strategizing targeted therapy approaches. Understanding the activation of metabolic signaling could also provide an opportunity to refine the strategies to disrupt tumor biology mediated by the IGF system. To further characterize the effects of the IGF system on metabolism, we looked at the real-time effect of IGF-1 and insulin on glycolytic ATP (glycoATP) and mitochondrial ATP (mitoATP) production. We observed an increase in glycolytic ATP production in MCF-7L cells when treated with IGF-1 at physiological conditions, implicating its role in increasing ATP production in BC cells via glycolysis.

## 2. Results

### 2.1. IGF-1 Increases ECAR Response of MCF-7L Cells

We were interested to see if treating MCF-7L cells with IGF-1 (5 nM), insulin (10 nM), or vehicle (Dulbecco’s modified Eagle medium (DMEM) supplemented with 10 mM glucose, 2 mM glutamine, and 1 mM sodium pyruvate) acutely would stimulate cellular metabolism. We examined oxygen consumption rate (OCR) and extracellular acidification rate (ECAR) (Figure 2A) upon acute injection of the ligands and recorded their response over 40 min. OCR is the measure of oxygen consumption that takes place during oxidative phosphorylation (OXPHOS)—a pathway contributing to ATP production. ECAR measures the acidification of assay media due to two ATP-producing pathways: glycolysis and citric acid cycle (TCA) (fuels electron transport chain/OXPHOS). Glycolysis releases one H+ per glucose to lactate conversion, and TCA releases CO_2_, both of which lead to acidification of the assay media. After ligand injection, IGF-1 treatment led to an increase in ECAR response, while insulin did not when compared to vehicle. Neither of the treatments could induce a change in the OCR readings (Figure 2B).

To further verify if the ECAR response with IGF-1 is via IGF-1R or IR, a MCF-7L cell line selected for tamoxifen resistance was evaluated. We have shown that tamoxifen-resistant MCF-7L (MCF-7L TamR) cells have lower levels of IGF-1R with unchanged levels of IR [45]. We hypothesized that IGF-1 induction should not elicit any ECAR response on this cell line if the signaling is taking place via IGF-1R. This was verified in our results shown in Figure 3. Neither IGF-1 nor insulin increased ECAR or OCR when compared with vehicle (Figure 3A,B). Since the results were as expected, we were able to verify that IGF-1/IGF-1R interaction generated an ECAR response in MCF-7L cells. These results warranted a need to quantify the independent contribution of glycolysis and oxidative phosphorylation in ATP production upon stimulation. Both ATP production pathways contribute to ECAR, but only oxidative phosphorylation contributes to OCR. Since an increase was observed only in ECAR, we speculated that IGF-1 induces an increased incidence of glycolysis in MCF-7L cells.

### 2.2. IGF-1 Upregulates Glycolysis in MCF-7L Cells

ECAR trends observed during the initial stimulation experiments speculated that IGF-1 increased ATP production in BC cells via glycolysis. To further confirm this finding, we used a real-time ATP rate assay to quantify the individual contributions of oxidative phosphorylation and glycolysis to total ATP generation. We replicated the previous experiment conditions by providing acute treatment of insulin (10 nM), IGF-1 (5 nM), and vehicle (DMEM media supplemented with 10 mM glucose, 2 mM glutamine, and 1 mM sodium pyruvate) to MCF-7L and MCF-7L TamR cells for 40 min. After which mitochondrial modulators—Oligomycin and Rotenone/Antimycin A (AA)—were injected, and data was recorded for 18 min each (Figure 4). Oligomycin blocks complex V on the mitochondrial membrane by inhibiting ATP synthase, thus decreasing the electron flow through ETC and, ultimately, mitochondrial respiration or OCR. This reduction in OCR is linked to cellular ATP production. Rotenone and Antimycin A are complex I and complex III inhibitors, respectively, which completely shut off mitochondrial respiration. Due to this, non-mitochondrial respiration can be quantified.

The Seahorse analytics platform was then used to plot mitochondrial ATP and glycolytic ATP production rate graphs, basal and induced ATP production rates, and energetic plots (Figure 5). These plots were generated according to calculations described in the materials and methods section. IGF-1 promoted glycolytic ATP production in MCF-7L cells with no change in mitochondrial ATP production when compared with vehicle (Figure 5A,B). This confirmed that IGF-1 promotes ATP production via glycolysis in MCF-7L cells. Insulin did not affect MCF-7L’s ATP generation rate as it remained constant and comparable to the vehicle group (Figure 5A,B). A 27.28% increase in glycolytic ATP production upon IGF-1 treatment in MCF-7L cells was quantified by comparing the basal and induced conditions (Figure 5C) for total ATP production. For MCF-7L TamR cells, neither produced a change in ATP generation rate when compared with the vehicle group (Figure 5A–C). Energetic plots (Figure 5D,E) also showed an increase in glycolytic ATP production in MCF-7L cells when treated with IGF-1, as depicted by the shift in induced vs. basal condition.

## 3. Discussion

In this paper, we studied the metabolic phenotype of BC cells in response to IGF system stimulation. For this purpose, we looked at the real-time ATP production rate in MCF-7L and MCF-7L TamR cells when stimulated with ligands—IGF-1 and insulin. Their effect was compared to a vehicle group which was treated with assay media. We were able to show a heightened glycolytic ATP production rate in MCF-7L cells when stimulated with IGF-1. In contrast, cells with low IGF-1R (MCF-7L TamR) did not increase ATP production with either IGF-1 or insulin. Since the ligand activity was tested at physiological concentrations, we can conclude two things—IGF-1 and IGF–1R interaction may be responsible for the increased metabolic activity of BC cells, and these cells rely on the glycolytic pathway to increase ATP production and meet the metabolic demand for proliferation.

These findings are consistent with the Warburg effect, which details aerobic glycolysis as a means of ATP generation in tumors [27]. Further evidence of IGF-1R’s role in regulating metabolism via the Warburg effect has also been shown in colorectal cancer [47]. While normal cells depend on the insulin-IR interaction for enhanced uptake and utilization of glucose, these cancer cells did not utilize IR for enhanced glycolysis. One reason could be that we tested insulin at physiological concentrations rather than supraphysiologic levels. In the BC environment, hyperinsulinemia is often observed, which probably translates to increased IR activity. Hyperinsulinemia has also been indicated to reduce IGFBPs production and increase IGF-1 biological activity, ultimately leading to cancer progression [24,48]. Investigating IR response in a high insulin environment as well as IGF-1 response in a high insulin environment can potentially provide meaningful insight into insulin’s role in metabolic dysregulation. A study conducted on IGF-1R knockout MCF7 cells with a high IR-A:IR-B ratio showed an amplified ECAR and OCR response upon 20h treatment with 10 nM insulin [26], thus indicating that insulin at physiological concentrations participates in metabolic regulation in a receptor-rich environment.

Several studies on the IR and its isoforms have shown IR-B to be involved predominantly in metabolism and IR-A in fetal growth and development with incidence in cancer progression [49,50,51,52]. A previous study by our group has shown inhibition in the growth of endocrine-resistant BC cells by IR disruption through blocking peptides or shRNA [40]. Targeting both isoforms of IR likely leads to physiological imbalance. More specific targeting of each isoform could potentially avoid this imbalance. The side effects of anti-IGF-1R therapy—hyperinsulinemia and hyperglycemia—warrant the further development of strategies to manage this toxicity. Ideally, targeting IR-A specifically, and leaving IR-B unaffected, might be an effective IGF system targeting approach. These studies can help bridge the gap in understanding the correlation between metabolic dysregulation, cancer progression, and the IGF axis.

Our data showed that glycolytic ATP production output increases in BC cells due to IGF-1. This finding implies that IGF-1/IGF-1R interaction assists BC cells in meeting the metabolic demand for proliferation via glycolysis. Further study into the synergistic effect of insulin and IGF-1 to investigate IGF-1 response in hyperinsulinemia conditions can help understand the correlation between metabolic dysregulation and the IGF axis in cancer progression.

## 4. Materials and Methods

### 4.1. Reagents

Growth media and supplements were purchased from Thermo Fisher Scientific, MA, USA. IGF-1 was purchased from Gemini, CA, USA. Insulin was purchased from Eli Lilly, IN, USA. Tamoxifen was purchased from Sigma Aldrich, MO, USA. Seahorse DMEM media (phenol red free), supplements (glucose, L-glutamine, sodium pyruvate), real-time ATP rate assay kit, and calibrant were purchased from Agilent Technologies. Hoechst dye was purchased from Thermo Fisher Scientific.

### 4.2. Cell and Cell Culture

Both cell lines were maintained at 37 °C in a humidified atmosphere containing 5% CO_2_. MCF-7L cells were from C. Kent Osborne (Baylor College of Medicine, Houston, TX, USA) and are routinely maintained in improved MEM Richter’s modification medium (zinc option) supplemented with 11.25 nmol/L insulin, 1% penicillin and streptomycin, and 5% fetal bovine serum. MCF-7L TamR cells were previously generated as described [45]. They are routinely maintained in phenol red free IMEM (zinc option) supplemented with 11.25 nmol/L insulin, 5% charcoal/dextran treated fetal bovine serum, and 100 nmol/L 4-OH tamoxifen.

### 4.3. Mitochondrial Bioenergetics

Agilent Seahorse XFe96 analyzer was used to study mitochondrial bioenergetics using the method described [53]. MCF-7L (1.2 × 10^4^ per well) and MCF-7L TamR (1.5 × 10^4^ per well) cells were seeded in Seahorse XF96 V3 PS cell culture microplates in their respective growth media. They were then washed with seahorse assay media (DMEM (pH 7.4; 103575-100; Agilent Technologies) + 10 mM glucose (1 M, 103577-100; Agilent Technologies) + 2 mM glutamine (200 mM, 103579-100; Agilent Technologies) + 1 mM sodium pyruvate (100 mM, 103578-100; Agilent Technologies)) twice; one hour before and just before running the assay. A real-time ATP rate assay kit was used to determine the contribution of mitochondrial respiration and glycolysis in ATP production. The assay uses two electron transport chain (ETC) modulators: Oligomycin and a mixture of Rotenone + Antimycin A (Rotenone/AA) to block ETC sequentially. ECAR and OCR readings upon modulator injections are then used to calculate basal and induced glycolytic ATP and mitochondrial ATP production rates, as indicated in the calculations section. Oligomycin blocks complex V on the mitochondrial membrane by inhibiting ATP synthase, thus decreasing the electron flow through ETC and, ultimately, mitochondrial respiration or OCR. This reduction in OCR is linked to cellular ATP production. Rotenone and Antimycin A are complex I and complex III inhibitors, respectively, which completely shut off mitochondrial respiration. Due to this, non-mitochondrial respiration can be quantified. After collecting the basal data, ligands—insulin (10 nM), IGF-1 (5 nM), and vehicle (Dulbecco’s modified Eagle medium (DMEM) supplemented with 10 mM glucose, 2 mM glutamine, and 1 mM sodium pyruvate)—were injected and their responses were recorded for 40 min. After this, oligomycin (2 µM) and rotenone/AA (0.5 µM) were serially injected, and their effect was recorded over 18 min via three measurements. The ATP production rate was normalized to per 1000 cells by counting the cells using brightfield and fluorescence images in BioTek cytation, Agilent Technologies. Hoechst dye was used at 2× concentration for staining.

### 4.4. Calculations

The data obtained were plotted using GraphPad PRISM 7, where the area under the ROC curve was calculated and compared to the control group to determine the significance of the treatment response. Data were also analyzed using Seahorse analytics by Agilent Technologies (https://www.agilent.com/en/product/cell-analysis/real-time-cell-metabolic-analysis/xf-software/agilent-seahorse-analytics-787485, accessed on 16 April 2023). Mitochondrial ATP Production Rate: The rate of ATP production (expressed in pmol ATP/min) associated with OXPHOS in the mitochondria. Basal rate calculation: (Last OCR rate measurement before first injection − Minimum OCR rate measurement after oligomycin but before Rotenone/Antimycin A injection) × 2 × (P/O). Induced rate calculation: [(Average OCR rate measurement after acute injection and before oligomycin injection) − Minimum OCR rate measurement after oligomycin and before Rotenone/Antimycin A injection) × 2 × (P/O). Glycolytic ATP Production Rate: The rate of ATP production (expressed in pmol ATP/min) associated with the conversion of glucose to lactate in the glycolytic pathway. Basal rate calculation: (Last glycoPER measurement before first injection). Induced rate calculation: (Average of the glycoATP Production Rate measurements after the acute injection and before the next injection).

### 4.5. Statistical Analysis

The area under the ROC curve for each group as well as the standard error was calculated using GraphPad PRISM as described [54]. These values were then input into the formula: (|Area1 − Area2|/√((SE2Area1) + (SE2Area2)) to give the value of z. The value of z was used to calculate the two-tail p-value by using the Microsoft Excel function: =2 × (1 − NORMSDIST(z)). For comparing induced rates to basal rates, the mean and standard deviation for treatment groups were normalized with the vehicle group and statistically analyzed using *t*-tests. A *p*-value < 0.05 was considered significant.

## Figures and Tables

**Figure 1 ijms-24-10209-f001:**
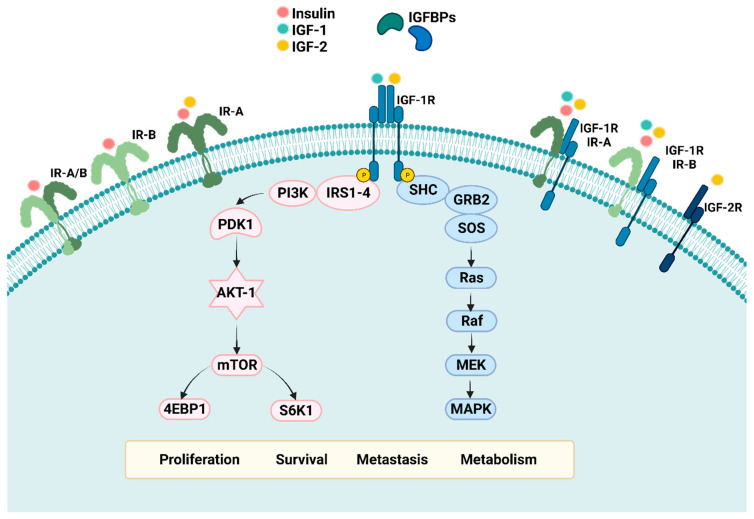
The IGF system and its components. The IGF family comprises several receptors, including transmembrane tyrosine kinase-linked receptors (RTKs)—Insulin-like growth factor receptor 1 (IGF-1R), Insulin-like growth factor receptor 2 (IGF-2R), Insulin receptor (IR) A and B, and their hybrids; three structurally and functionally similar ligands—IGF-1, IGF-2, and Insulin; and six soluble binding proteins: Insulin-like growth factor binding proteins (IGFBPs) 1–6. Depending on the target tissue, the levels of the binding proteins are regulated by IGFBP proteases (not shown). Receptor ligand interaction leads to downstream signaling through PI3K-AKT and RAS-RAF-MAPK, which regulates proliferation, survival, metastasis, and metabolism. Created with BioRender.com.

**Figure 2 ijms-24-10209-f002:**
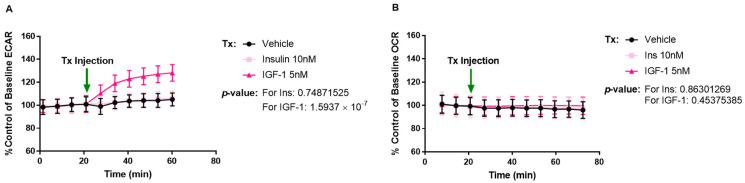
IGF-1 enhances the ECAR response of MCF-7L cells. MCF-7L cells were analyzed using the Seahorse XFe96 analyzer, Agilent Technologies, CA, USA, and the OCR and ECAR readings were recorded. Readings were normalized to per 1000 cells by using fluorescence imaging. Tx indicates 5 nM IGF-1, 10 nM insulin, and Vehicle injection (Seahorse DMEM media supplemented with glucose (10 mM), L-glutamine (2 mM), and sodium pyruvate (1 mM)). (**A**) There was a gradual increase in the ECAR readings when stimulated with 5 nM IGF-1 as compared to vehicle (Seahorse DMEM media supplemented with glucose (10 mM), L-glutamine (2 mM), and sodium pyruvate (1 mM)) (*p*-value ≤ 1.5937 × 10^−7^) but not with 10 nM insulin stimulation. (**B**) MCF-7L cells did not show any change in OCR upon ligand stimulation. A *p*-value < 0.05 was considered significant.

**Figure 3 ijms-24-10209-f003:**
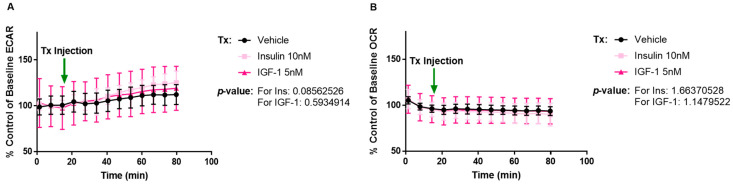
IGF-1 or Insulin do not enhance ECAR or OCR response in MCF-7L TamR cells. Tx indicates 5 nM IGF-1, 10 nM insulin, and Vehicle injection (Seahorse DMEM media supplemented with glucose (10 mM), L-glutamine (2 mM), and sodium pyruvate (1 mM)). (**A**,**B**) MCF-7L TamR cells were analyzed using the Seahorse Xfe96 analyzer. No change was observed in the ECAR and OCR readings when these cells were stimulated with 5 nM IGF-1 and 10 nM insulin as compared to vehicle (Seahorse DMEM media supplemented with glucose (10 mM), L-glutamine (2 mM), and sodium pyruvate (1 mM)). A *p*-value < 0.05 was considered significant.

**Figure 4 ijms-24-10209-f004:**
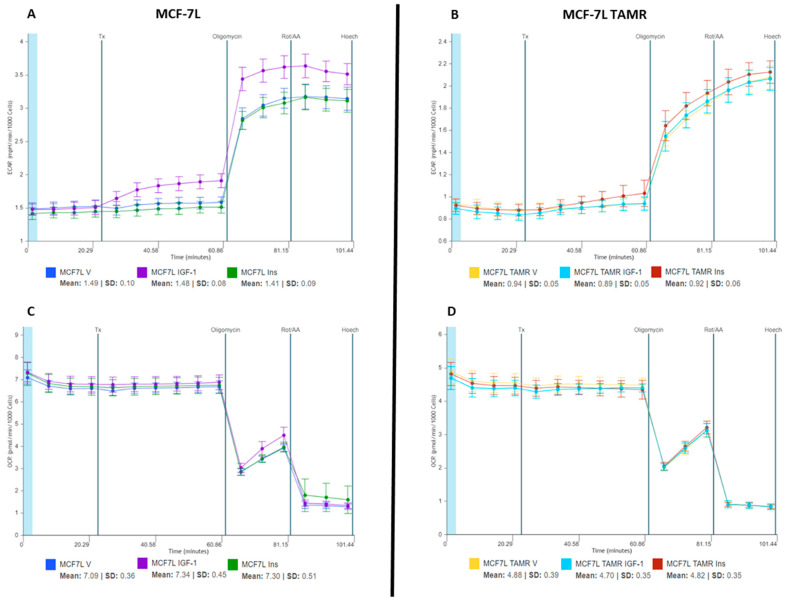
The ECAR and OCR responses of MCF-7L and MCF-7L TamR cells to ligand and inhibitor treatments. Readings were normalized to per 1000 cells by using fluorescence imaging. Tx indicates 5 nM IGF-1, 10 nM insulin, and vehicle injection (Seahorse DMEM media supplemented with glucose (10 mM), L-glutamine (2 mM), and sodium pyruvate (1 mM)). (**A**,**C**) For MCF-7L cells, the ECAR and OCR results were replicated with the ligand response—10 nM IGF-1 (increase) and 5 nM insulin (no change) as compared to vehicle (Seahorse DMEM media supplemented with glucose (10 mM), L-glutamine (2 mM), and sodium pyruvate (1 mM)) (*p*-value for IGF-1 response compared to vehicle ≤ 1.0386 × 10^−9^). The dip in OCR readings and the increase in ECAR readings with Oligomycin and Rotenone/AA injection are in line with the inhibitor effect. (**B**,**D**) For MCF-7L TamR cells, the ECAR and OCR results were replicated with the ligand response—10 nM IGF-1 (no change) and 5 nM insulin (no change) as compared to vehicle (Seahorse DMEM media supplemented with glucose (10 mM), L-glutamine (2 mM), and sodium pyruvate (1 mM)). The dip in OCR readings and the increase in ECAR readings with Oligomycin and Rotenone/AA injection are in line with the inhibitor effect. A *p*-value < 0.05 was considered significant.

**Figure 5 ijms-24-10209-f005:**
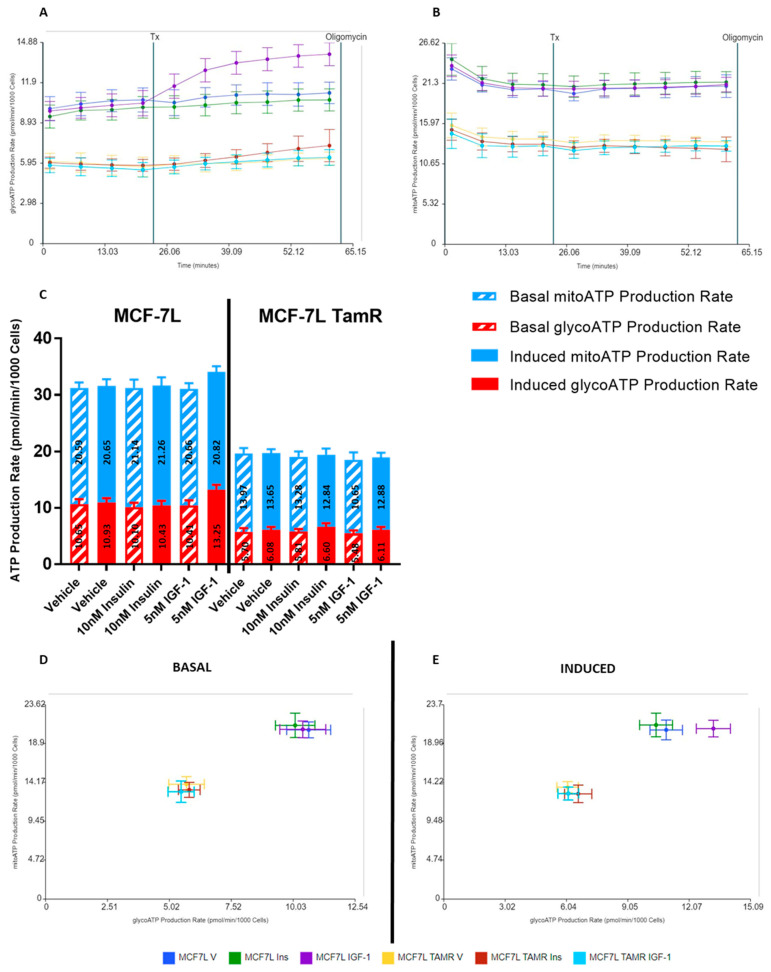
Glycolytic ATP production rate increases in MCF-7L cells when stimulated with IGF-1. Readings were normalized to per 1000 cells by using fluorescence imaging. Tx indicates 5 nM IGF-1, 10 nM insulin, and vehicle injection (Seahorse DMEM media supplemented with glucose (10 mM), L-glutamine (2 mM), and sodium pyruvate (1 mM)). (**A**) MCF-7L cells showed an increase in glycolytic ATP production after 5 nM IGF-1 injection (*p*-value ≤ 1.9418 × 10^−10^), but no change was observed after 10 nM insulin injection as compared to vehicle (Seahorse DMEM media supplemented with glucose (10 mM), L-glutamine (2 mM), and sodium pyruvate (1 mM)). For MCF-7L TamR cells, there was no apparent change with either treatment. (**B**) mitochondrial ATP production rate remained constant after treatment injection, indicating no response. (**C**) There was an increase in the total ATP production for the MCF-7L group after being induced with IGF-1, while other groups did not show any change as compared to basal levels. This increase is largely due to the glycolytic ATP production rate (*p*-value ≤ 0.0000003). (**D**,**E**) The energetic maps further give evidence of the increase in glycolytic ATP production by MCF-7L cells when induced with 5 nM IGF-1 as compared to basal conditions. This is depicted by the shift towards glycolysis after IGF-1 treatment (*p*-value ≤ 0000013). At the same time, insulin shows no effect on both cell lines.

## Data Availability

Original data are available by request.

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
