# Peer review of "IGF-1 Stimulates Glycolytic ATP Production in MCF-7L Cells"

_ijms, 2023, doi:10.3390/ijms241210209_

Round 1
Reviewer 1 Report
This manuscript aims to characterize how metabolism is regulated by the IGF-1/insulin signaling pathways in breast cancer and is motivated by the need define metabolic regulation in breast cancer to better understand why treatments targeting the IGF-1 signaling pathway have not been successful for breast cancer. There are several points to address to improve the quality of this contribution.
1. Data for figure 4 is missing and must be included.
2. There is a section describing how data were statistically analyzed yet there are not statistics provided for any figure making interpretation difficult.
3. MCF-7L TamR cells are used to demonstrate the requirement for IGF-1R in the observed IGF-1-dependent metabolic regulation. This analysis can be improved by also inhibiting this signaling pathway by any of the inhibitors mentioned (references 36-40).
4. Methods other than seahorse analysis such as glucose uptake reporters could should also be used in experiments with similarly treated cells to show metabolic regulation by IGF-1 to further support the hypothesis.
Author Response
- Data for figure 4 is missing and must be included. We apologize for the inadvertent omission of this figure, it is now included in the revision.
- There is a section describing how data were statistically analyzed yet there are not statistics provided for any figure making interpretation difficult. The statistical analyses of the figures are now more clearly shown.
- MCF-7L TamR cells are used to demonstrate the requirement for IGF-1R in the observed IGF-1-dependent metabolic regulation. This analysis can be improved by also inhibiting this signaling pathway by any of the inhibitors mentioned (references 36-40). We thank the reviewer for making this suggestion. As noted in the response to this reviewer’s query #4, we are very interested in the signaling pathway and fuel source for the findings made in this paper. As we have previously shown, IGF signaling upregulates the transporter (SLC7A11 – xCT) to effect glutamine/glutathione metabolism and it is possible that either glutamine or glucose, or both, account for the phenotype we show in this work (Yang Y, Yee D. IGF-I regulates redox status in breast cancer cells by activating the amino acid transport molecule xc-. Cancer Res 74:2295-2305 2014. PMC ID: 4006361). Thus, we are planning a more comprehensive analysis of the effects of the inhibitors and fuel source. We feel that this more extensive evaluation of the signaling pathways and fuel source is outside of the scope of this work.
- Methods other than seahorse analysis such as glucose uptake reporters could/should also be used in experiments with similarly treated cells to show metabolic regulation by IGF-1 to further support the hypothesis. As mentioned above, we are also interested in understanding the fuel source for the findings. We have previously evaluated glucose uptake after IGF-1R downregulation (Zhang H, Pelzer AM, Kiang DT, Yee D. Down-regulation of type I insulin-like growth factor receptor increases sensitivity of breast cancer cells to insulin. Cancer Res 67:391-397 2007), but as mentioned above, we would like to do a more comprehensive analysis of potential metabolites. We feel that these additional experiments are outside the scope of this work.
Author Response
- First two introduction sentences need references. These references were added.
- In figure one it should to indicate that if it is no binding proteases degrade through IGFB1-7 by proteases. We can clarify in the figure legend that IGFBP proteases exist to influence levels of the binding proteins.
- Is needed to indicate pValue of any significances in figure 5. The p values have been added.
- You need to answer what is the relevance, what is the novelty depending on no too old published data in your topic, for example the same has been found in colorectal cancer: 10.3390/ijms22126434. We thank the reviewer for bringing this article to our attention. It further reinforces the concept that IGF-1R may have in regulating metabolism in cancer. The paper cited by the reviewer is now mentioned in the Discussion. Of note, this paper in colorectal cancer served as a review and didn’t provide experimental evidence of the differences between IGF-I and insulin. We believe our paper contributes to the field showing the role for IGF-1 in breast cancer regulation of metabolism while insulin has a lesser effect.
Round 2
Reviewer 1 Report
Previous oncerns were adequately addressed.